# Synergistic reduction of graphene oxide using vitamin C and urea: Enhanced efficiency and material properties

Fei-hu Zeng[1¤a*], SyYi Sim[2¤b*], Zhi-wen Wang[1], Feng Wang[1]

**1** Liming Vocational University, Quanzhou, Fujian, China, **2** Faculty of Engineering Technology, Universiti Tun Hussein Onn Malaysia, Pagoh, Johor, Malaysia

¤a Current Address: Liming Vocational University, Quanzhou, Fujian, China
¤b Current Address: Faculty of Engineering Technology, Universiti Tun Hussein Onn Malaysia, Parit Raja, Batu Pahat, Johor, Malaysia
* sysim@uthm.edu.my (SS), 13859765695@163.com (FZ)

## Abstract

Synergistic reduction of graphene oxide (GO) using different reducing agents represents an effective approach for reduced graphene oxide (rGO) synthesis. In this study, the rGO (rGO-Vc+Urea) was prepared by combining vitamin C (Vc) and urea as co-reducing agents with the modified Hummer's method. Compared to samples reduced solely with Vc or urea, the co-reducing agents significantly reduced the required reaction time (to 2 hours) and temperature (to 120°C), while yielding material with superior electrical resistivity (1.2 Ω·cm). The structure of the samples was characterized using XRD, FT-IR, Raman spectroscopy, BET surface area analysis, and SEM. Results indicate that the sample prepared from co-reducing agents possesses a typical graphene structure and incorporates C-N bonds. Furthermore, rGO-Vc+Urea exhibits a higher degree of structural order, as evidenced by a lower Raman ($I_D/I_G = 0.75$), compared to rGO-Vc ($I_D/I_G = 0.91$) and rGO-Urea ($I_D/I_G = 1.49$), along with a higher specific surface area (88.60 m²/g). The reduction mechanism of the co-reducing agents was investigated. It was revealed that the alkaline environment generated by urea enhances Vc's ability to reduce oxygen-containing functional groups in GO, specifically hydroxyl, epoxy, carbonyl, and carboxyl groups, and promotes the elimination of $CO_2$ released during the reaction. This strategy of employing synergistic multiple reducing agents offers new perspectives for the preparation of rGO.

## Introduction

Graphene, a two-dimensional carbon material with conjugated sp² hybridization, exhibits exceptional mechanical strength, electrical conductivity, thermal conductivity, and carrier mobility in its single-layer form [1]. Consequently, it finds applications in diverse fields, including optoelectronic devices [2], sensors [3], biomedicine [4],

**Data availability statement:** All relevant data are within the paper and its Supporting Information files.

**Funding:** This work was supported by the Science and Technology Program (LT202103) and Innovation Team Program of Liming Vocational University. The funders had no role in study design, data collection and analysis, decision to publish, or preparation of the manuscript.

**Competing interests:** The authors have declared that no competing interests exist.

energy storage (including batteries) [5,6], electrochemical systems [7], thermal management [8], and armor materials [9]. Numerous methods exist for graphene preparation, broadly categorized into top-down approaches (such as exfoliation [10,11], ultrasonication [12], ball milling [13], and peeling [14]) and bottom-up approaches (including liquid-phase exfoliation, electrochemical reduction [15], and reduction of graphite oxide [16–19]).

The reduction method offers several advantages, including high yield, low cost, and ease of chemical modification. Unlike pristine graphene, composed entirely of $sp^2$-hybridized carbon structures, rGO contains $sp^2$-hybridized regions alongside $sp^3$ hybridized domains and defects formed during epitaxial growth in the reduction process. Additionally, rGO may retain residual oxygen-containing functional groups inherited from its precursor GO [18]. This structural complexity arises from the assembly of graphene into nanoflake structures with heterogeneous heights and diameters, which aggregate into varied configurations during GO reduction [19]. The interlayer spacing of rGO is highly dependent on the reduction degree. Consequently, the electrical conductivity of rGO is influenced not only by the reducing agent type but also by morphological characteristics of the resultant and layers, such as flake size, defect type, and defect density within the graphene lattice.

A wide variety of substances serve as reducing agents for preparing rGO. Strong reducing agents including hydrazine and its derivatives (e.g., hydrazine hydrate [20], dimethylhydrazine [21]), metal hydrides (e.g., sodium hydride [22], sodium borohydride [23]), and hydroiodic acid [24]. Weak reducing agents encompass Vc [8], strongly alkaline solutions (e.g., KOH [25], NaOH [26]), hydroxylamine [27], and urea [16]. Galvao et al. [28]compared the effectiveness of four co-reductant systems for GO reduction in GO synthesis: rGO: $Vc + H_2O_2$, sodium hydrosulfite ($Na_2S_2O_4$) + NaOH, inorganic salts + NaOH, and polysaccharides + NaOH, where NaOH primarily functioned as a pH regulator. Gao [29] reported an environment-friendly methodutilizing Vc as the reductant and amino acid as a stabilizer. These studies demonstrated the efficacy of co-reductants in facilitating reduction. Given the current prioritization of safety and environmental sustainability, Vc and urea have garnered significant attention as reducing agents. However, their reduction mechanisms remain incompletely understood, and a universal mechanisms explanation is lacking. Vc possesses a unique molecular structure featuring a highly reactive enediol group, conferring strong reducing power due to its high propensity for electron donation [30]. In contrast, urea not only directly reduces GO but also decomposes to release alkaline species [31]. This ammonia-derived alkalinity enhances the electron-donating capability of Vc, thereby improving its reducing power for efficient GO reduction. Comparing the reduction performance of Vc and urea individually, as well as combining them as co-reductants, provides valuable insights into the reaction mechanisms governing rGO preparation from GO.

In this study, rGO was prepared through the reduction of GO, synthesized via the modified Hummers' method, using Vc, urea, or their mixture as co-reducing agents. The structure of the resulting rGO samples was characterized and comparatively analyzed using X-ray diffracton(XRD), Fourier-transform infrared

spectroscopy(FT-IR), Raman spectroscopy, Brunauer-Emmett-Teller(BET) surface area analysis, and scanning electron microscopy(SEM). The electrical conductivity of the rGO samples was also assessed and compared. Furthermore, the reduction mechanisms of both Vc and urea were elucidated and compared in detail.

## Experimental section

### Material and chemicals

Flake graphite (100 mesh, ~200 µm particle size) was purchased from Qingdao Dongkai Graphite Co., Ltd. (Qingdao, China). All other chemicals were obtained from Sinopharm Chemical Reagent Co., Ltd. (Shanghai, China). Detailed specifications are provided in Supporting Information (S1Table).

### rGO preparation

GO was synthesized using the modified Hummer's method [32], rGO preparaed using Vc as the reduced agent was denoted as rGO-Vc. For rGO-Vc synthesis, a GO film (0.3 g) was dispersed in 75 mL of deionized water via ultrasonication for 10 minutes. Vc was added at a specified Vc/GO mass ratio, and the reaction proceeded in a 100 mL reactor under controlled temperature and time conditions. The resultant product underwent repeated washing and centrifugation cycles with deionized water to remove residual Vc and reaction byproducts. Finally, rGO-Vc was isolated by vacuum freeze-drying. Similarly, samples reduced using urea and the Vc/urea mixture as reducing agent were denoted as rGO-Urea and rGO-Vc+Urea, respectively. All syntheses were performed in duplicate or triplicate.

### Characterization techniques

The electrical resistivity ($\rho$) and resistance (R) of the samples were analyzed using an ST2258C multifunctional digital four-point probe tester (Jingge Electronic Co., Ltd., Jiangsu, China). Structural changes during the reduction of GO to rGO were characterized by X-ray diffraction (XRD) on a MiniFlex600 diffractometer (Rigaku, Japan) using Cu K$\alpha$ radiation ($\lambda$ = 0.154 nm) at 40 kV, scanning the 2$\theta$ range from 10° to 50°. Alterations in chemical bonding and functional groups after reduction were examined by Fourier transform infrared spectroscopy (FT-IR) using an IRAffinity-1S spectrophotometer (Shimadzu, Japan). The microstructure of GO and rGO was analyzed using a LabRAM HR Evolution Raman spectrometer (Horiba Scientific, France). Specific surface areas were determined from nitrogen adsorption isotherms measured at 77 K, applying the Brunauer-Emmett-Teller (BET) model to the linear region of the adsorption curve (relative pressure $P/P_0$ = 0.05–0.30). Average pore size was evaluated using nonlocal density functional theory (NLDFT) analysis on a JW-BK200B analyzer (Beijing Jingwei Gaobo Technology Co., Ltd., China). Morphological characteristics of GO and rGO were examined by scanning electron microscopy (SEM) using a Nova200 instrument (FEI Company, USA) operated at an accelerating voltage of 10.0 kV.

## Results and discussion

### Conductivity

Building on previous studies [33,34], we evaluated the electrical resistivity of rGO synthesized using Vc and urea as reducing agents. An optimal reaction temperature is critical for effective reduction (Fig 1a). rGO-Vc synthesized at 60°C exhibits high resistivity ($\rho$ = 55,700 $\Omega$·cm) and resistance (R = 79,000 $\Omega$), consistent with non-conductive behavior. Increasing the temperature to 100°C dramatically reduces these values to $\rho$ = 1.82 $\Omega$·cm and R = 2.75 $\Omega$, respectively—representing a decrease of over four orders of magnitude in resistivity. In contrast, rGO-Urea synthesized at 100°C shows significantly higher resistivity ($\rho$ = 40.45 $\Omega$·cm) and resistance (R = 50.85 $\Omega$), Comparable conductivity to rGO-Vc (100°C) was only achieved with urea at 180°C. These results demonstrate that urea requires substantially higher reduction temperatures than Vc to attain equivalent electrical properties.

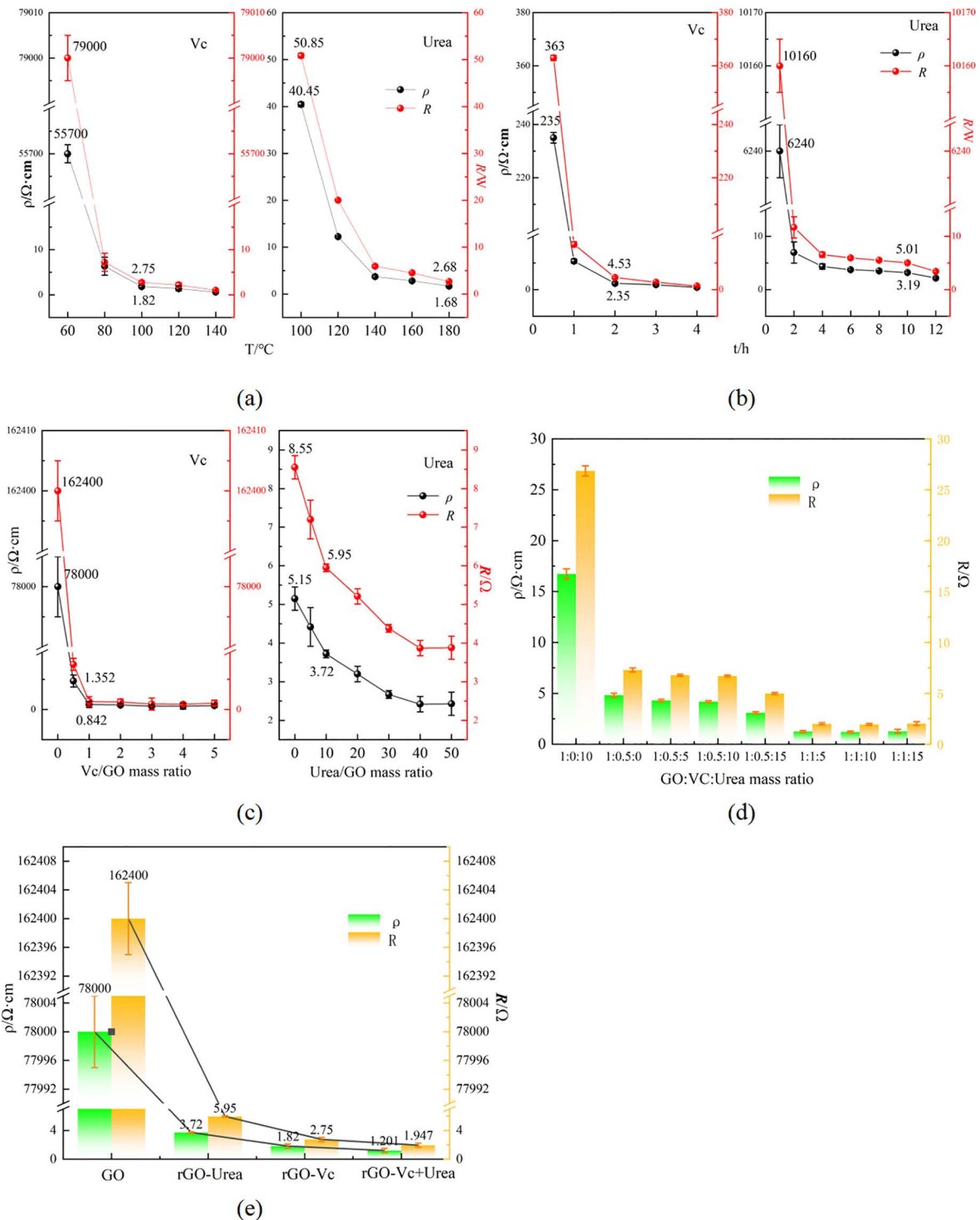

**Fig 1. Conductivity of samples((a) Conductivity of rGO-VC and rGO-Urea at different temperatures, (b)Conductivity of rGO-Vc and rGO-Urea at different time, (c) Conductivity of rGO-Vc and rGO-Urea at different mass ratio, (d) Conductivity of rGO-Vc + Urea(120°C, 2h) at different mass ratio, (e) Conductivity of GO, rGO-Vc(Vc/GO mass ratio of 1 to 1,100°C,3h), rGO-Urea(Urea/GO mass ratio of 10 to 1,140°C,4h)and rGO-VC + Urea mass ratio of GO(10):Vc(0.5):Urea(10), 120°C, 2 h)).**

The performance difference stems from distinct reduction mechanisms. Urea must thermally decompose to release nucleophilic amino groups (–NH$_2$), which attack epoxy groups (C–O–C) on GO, initiating ring-opening. This process forms hydroxyl intermediates that undergo dehydration to yield C=C bonds [35], Concurrently, hydrolysis-derived ammonia and residual diamide-configured urea molecules form nitrogen-doped heterocycles (e.g., pyrrolic/graphitic nitrogen) due to structural instability and carbonization [27]. In contrast, Vc reduces GO via direct electron transfer and proton donation [36]. Its weak acidity and high solubility enable rapid generation of ascorbate ions (C$_6$H$_7$O$_6^-$) and protons (H$^+$), efficiently reducing oxygen function groups. Consequently, Vc achieves substantial conductivity within 2 hours, while urea reduction requires up to 10 hours (Fig 1b) due to slow accumulation of active species under hydrothermal conditions. This kinetic disparity correlates with the spatially heterogeneous reduction of sp$^3$-type defects, governed by reductant-specific properties [19].

Reductant dosage further highlights efficiency differences between Vc and urea. Hydrothermal treatment alone (100–140°C) achieves only partial GO reduction [37]. With Vc at a Vc/GO mass ratio of 1:1, resistivity drops to 0.842 Ω·cm (R = 1.352 Ω). In contrast, Urea requires a 10:1 mass ratio to attain 3.72 Ω·cm (R = 5.95 Ω). Pristine GO exhibits exceptionally high resistivity (ρ = 78,000 Ω·cm, R = 162,400 Ω; Fig 1e), reflecting extensive oxidation where intercalated oxygen disrupts the π-conjugated network via functional groups (e.g., C–O, C=O), significantly enlarge interlayer spacing, and degrades conductivity [27].

The synergistic combination of VC and urea significantly reduces the electrical resistivity of rGO (Fig. 1d) while simultaneously lowering the reduction temperature and shortening reaction duration compared to either reductant alone (Fig 1e). Hydrothermal hydrolysis of urea generates an alkaline environment that enhances electron transfer efficiency. This alkaline milieu promotes deprotonation of Vc (C$_6$H$_7$O$_6^-$), which efficiently donates electrons to epoxy and carbonyl groups on GO while suppressing oxidative side reactions of Vc. Concurrently, urea facilitates the removal of recalcitrant carboxyl groups at elevated temperatures. The resulting nitrogen-doped graphene exhibits enhanced electronic properties and structural versatility due to the incorporation of diverse nitrogen configurations within the carbon lattice [38].

## XRD analysis

Structural evolution from GO to rGO was characterized by X-ray diffraction (Fig 2). GO exhibits a prominent (001) diffraction peak at 2θ ≈ 11° and a minor (100) peak at 42°, corresponding to an interlayer spacing of 0.773 nm [29]. After reduction, rGO-Vc, rGO-Urea and rGO-Vc+Urea exhibit broad (002) peaks centered around 25.1° and 25.6°, with interplanar spacings of 0.354 nm, 0.355 nm and 0.348nm, respectively. These peak broadening indicates a short-range order in the crystalline structure, and the decreased d-spacing confirms removal of oxygen functional groups and partial restoration of the graphitic network [5]. For rGO-Urea, a small peak persists near 42°, indicating residual C-O groups and/or C-N bond formation from nitrogen doping. Additionally, two minor characteristic GO peaks observed at 11° and 13° [39] indicate incomplete reduction of the rGO-Urea. Compared to rGO-Urea. the XRD peaks of rGO-Vc and rGO-Vc+Urea showed broadened peaks, further suggesting smaller crystal grains or greater micro strain within the crystal lattice [40].

## FT-IR analysis

FT-IR spectra of samples are shown in Fig 3. Broad peaks within 3100–3600 cm$^{-1}$ correspond to O–H stretching vibrations from adsorbed water. For GO, characteristic peaks at 1732 cm$^{-1}$ (C=O stretching), 1377 cm$^{-1}$ (C–OH deformation), and 1062 cm$^{-1}$ (C–O stretching) confirm the presence of carboxylic acid and carbonyl groups [41]. The aromatic C=C stretching vibration appears at 1624 cm$^{-1}$ [42]. The disappearance of these peaks in the reduced samples confirms the effective reduction of GO. Compared to GO, the C=O bond vibration peak at 1724 cm$^{-1}$ substantially diminishes after reduction, indicating that C=O bonds on the GO surface were fully reduced. Significant decreases in peaks at 1375 cm$^{-1}$ and 1061 cm$^{-1}$ indicate that most oxygen-containing functional groups in GO were reduced. This observation aligns with the XRD analysis results. The peak at 1541 cm$^{-1}$ [29], attributed to the in-plane C=C skeleton vibrations and adsorbed phytochemicals(primarily carboxylate moieties from Vc and urea), is absent in GO. According to previous literature and FTIR analysis,

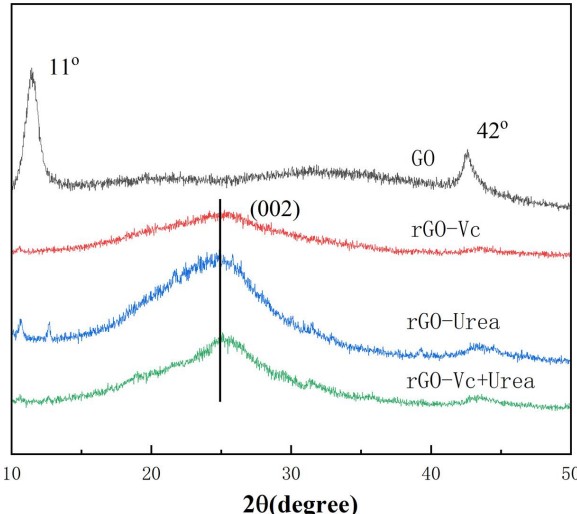

**Fig 2. XRD of GO, rGO-Vc, rGO-Urea and rGO-Vc+Urea(rGO-Vc(Vc/GO mass ratio of 1 to 1,100°C,3h), rGO-Urea(Urea/GO mass ratio of 10 to 1, 140°C, 4h) and rGO-Vc+Urea (mass ratio of GO(10):Vc(0.5):Urea(10)), 120°C, 2h)).**

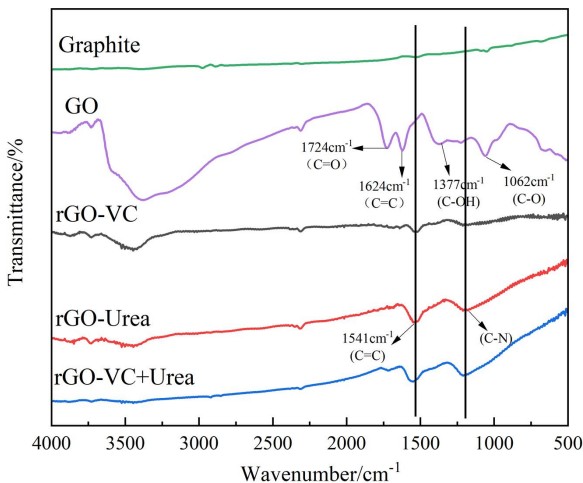

**Fig 3. FT-IR spectra of GO, rGO-Vc,rGO-Urea and rGO-Vc+Urea (rGO-Vc(Vc/GO mass ratio of 1 to 1,100°C,3h), rGO-Urea(Urea/GO mass ratio of 10 to 1,140°C,4h)and rGO-Vc + Urea (mass ratio of GO(10):Vc(0.5):Urea(10)), 120°C, 2h).**

hydroxyl and epoxide groups on the GO basal planes were removed via reaction with hydrogen atoms from Vc through $S_N2$ nucleophilic attack [43]. Additionally, samples reduced with urea exhibit a distinct peak cluster within the 1000–1300 $cm^{-1}$ range, which is absent in samples reduced with Vc. This cluster corresponds to C-N bond stretching vibrations resulting from nitrogen doping by urea [44].

## Raman analysis

The Raman spectra of GO, rGO-Vc, rGO-Urea and rGO-Vc+Urea are shown in Fig 4. In carbon materials, the observed G and D bands correspond to the $E_2g$ phonon mode of $sp^2$ carbon atoms and the breathing mode of ƙ-point phonons with

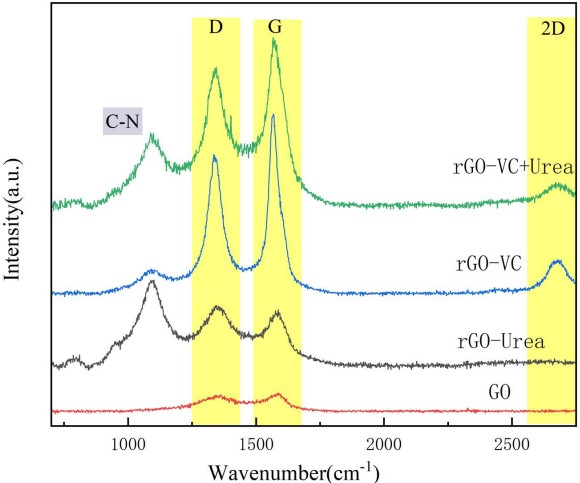

**Fig 4. Raman of GO, rGO-Vc, rGO-Urea and rGO-Vc+Urea(rGO-Vc(Vc/GO mass ratio of 1 to 1,100°C,3h), rGO-Urea(Urea/GO mass ratio of 10 to 1,140°C,4h)and rGO-Vc+Urea (mass ratio of GO(10): Vc(0.5): Urea(10)), 120°C, 2h)).**

$A_1g$ symmetry, respectively [45]. For GO in this study, the D and G bands appear at 1363 cm$^{-1}$ and 1591 cm$^{-1}$, yielding an intensity ration ($I_D/I_G$) of 0.96. Reduction of GO induces shifts in both the D and G band positions and changes their relative intensities. Specifically, the D bands shift to 1336 cm$^{-1}$ for rGO-Vc, 1342 cm$^{-1}$ for rGO-Urea and 1344 cm$^{-1}$ for rGO-Vc+Urea, indicating significant redshifts. The G bands are located at 1568 cm$^{-1}$ (rGO-Vc), 1590 cm$^{-1}$ (rGO-Urea) and 1567 cm$^{-1}$(rGO-Vc+Urea). Notably, the G band of rGO-Vc and rGO-Vc+Urea exhibits redshift, while no significant shift is observed for rGO-Urea. The G band redshift suggests a reduction in the number of graphene layers following reduction [46]. The calculated $I_D/I_G$ ratios are 0.91 for rGO-Vc, 1.49 for rGO-Urea, and 0.75 for rGO-Vc+Urea. This result aligns with findings by Brendan et al. [34] and Pankaj Chamoli et al. [39], indicating that rGO-Urea possesses lower structural order and a higher defect density compared to rGO-Vc and rGO-Vc+Urea. A higher defect density correlates with an increased number of exposed active carbon atoms on the graphene surface [47]. Additionally, rGO-Vc exhibited a prominent 2D peak at approximately 2700 cm$^{-1}$. This peak is attributed to the second-order zone-boundary phonons generated through double-resonance Raman scattering involving two-phonon emission [48]. The width of this peak suggests that the obtained rGO-Vc consists of multilayered graphene [49].

Furthermore, a peak at approximately 1090 cm$^{-1}$ is observed. This peak is commonly attributed to residual oxygen-containing functional groups (e.g., C-O) or nitrogen-related groups introduced during reduction. Its absence in the GO spectrum suggests it likely originates from residual Vc or nitrogen doping derived from urea. As a nitrogen-containing reducing agent, urea can form C-N bonds or other nitrogen functionalities onto the graphene surface, contributing to this peak. This observation aligns with findings indicating that urea reduction partially modifies the surface chemistry of graphene oxide, leaving detectable functional group signatures. This mechanism also accounts for the incorporation of nitrogen in rGO-Vc+Urea.

## BET analysis

BET curves of samples were shown in Fig 5. All three rGO materials exhibited Type II isotherms with H3 hysteresis loops [50], indicative of slit-shaped mesopores formed by the stacking of sheet-like particles (graphene sheets) and likely containing macropores. The relatively gentle slope in the low-pressure region suggests a minor contribution from micropores. Chemical modifying agents significantly influenced the pore structure. Co-modification with Vc + Urea (rGO-Vc+Urea)

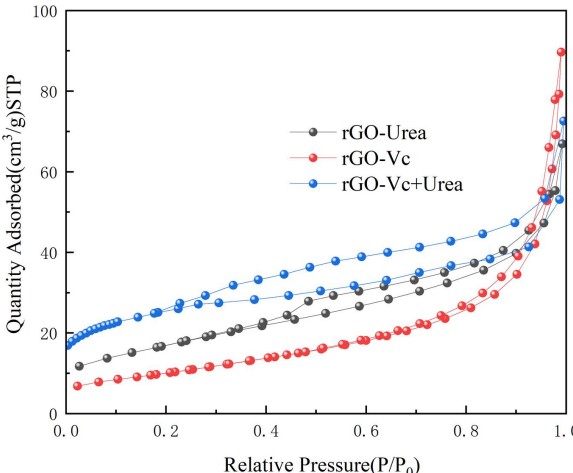

**Fig 5. BET curves of Samples(rGO-Vc(Vc/GO mass ratio of 1 to 1,100°C,3h), rGO-Urea(Urea/GO mass ratio of 10 to 1,140°C,4h)and rGO-Vc+Urea (mass ratio of GO(10): Vc(0.5): Urea(10)), 120°C, 2 h)).**

yielded the highest specific surface area(88.60 m$^2$/g), demonstrating a synergistic enhancement compared to modification with either Urea alone (rGO-Urea, 61.40 m$^2$/g) or Vc alone (rGO-Vc, 36.88 m$^2$/g). Among samples modified with a single agent, rGO-Urea exhibited marginally higher adsorption capacity and a slightly larger hysteresis loop than rGO -Vc, suggesting Urea may be more effective than Vc alone in mitigating graphene sheet restacking or generating additional void space. The mechanism likely involves urea decomposition during heating, releasing and reducing gases. These gases remove oxygen-containing groups and physically expand sp$^3$ defects in the rGO structure due to gas evolution [45]. When both agents are employed concurrently, a broader range of oxygen-containing functional groups is removed more completely. This extensive deoxygenation significantly reduces polar groups on the graphene sheets and markedly weakens interlayer hydrogen bonding. The presence of VC accelerates the reduction kinetics. It is plausible that this acceleration facilitates partial sheet reduction and imparts sufficient flexibility by the time gaseous products from urea decomposition are generated. This flexible, partially reduced state allows the sheets to be more readily expanded by the evolving gas bubbles, promoting the formation of a porous structure. Conversely, when urea is used alone, the lower reduction efficiency likely leaves the sheets insufficiently reduced and rigid, hindering effective pore creation by the bubbling action. Urea, also contributes significantly to dispersion stability through physisorption and the establishment of a hydrogen bonding network, providing enhanced steric hindrance. Owing to its higher specific surface area and abundant mesoporous structure, rGO-Vc+Urea is anticipated to exhibit enhanced performance over rGO-Vc and rGO-Urea in applications requiring high surface area and efficient mass transport, such as adsorption (gases, pollutants), catalyst supports, and specific supercapacitor electrodes. Key findings from related research using Vc and urea as reducing agents are summarized in Table 1, highlighting the advantages of their combination as a co-reducing agent for the preparation of rGO.

**SEM.** Aggregation and crumpling observed in the reduced samples are morphological indicators of successful reduction. Thermal vibrations and interatomic interactions induce spatial variations within sp$^2$-bonded carbon domains, causing the two-dimensional sheets to extend into three-dimensional space and ultimately generating increased rippling and folding. The SEM image of GO (Fig 6a) reveals a surface with a ridge-like appearance characterized by wrinkles and flake-like structures. In contrast, Figs 6b and 6c display aggregated nanosheets and twisted, stacked folds in the reduced samples. These morphological changes fundamentally arise from the reconstruction and reassembly of graphene sheets concurrent with the removal of oxygen-containing functional groups. Consequently, rGO exhibits a higher density of defects compared to GO. Notably, the edges of rGO-Vc (Fig 6b, c) show a higher apparent degree of crumpling than

**Table 1. Comparison of reported results on Vc and urea as reducing agents.**

| Reduce agent | T(°C) | t (h) | d spacing (nm) | Id/Ig | rGO resistance, conductivity or capacitance | Surface area | Average pore size | Ref. |
|---|---|---|---|---|---|---|---|---|
| lemon | 180 | 6 | 0.356 | – | – | – | – | [51] |
| Vc+Amino Acid | 80 | 24 | 0.34-0.37 | 0.98-1.17 | 14.1 S·m$^{-1}$ | – | – | [29] |
| Vc | 150 | 0.25 | – | 0.854 | 44 S·cm$^{-1}$ | – | – | [34] |
| Urea | 160 | 3 | 0.362 | – | 184.5F·g$^{-1}$ | 173.1m$^2$/g | 3-4nm | [33] |
| Urea | 170 | 12 | – | 0.9-1.06 | – | – | – | [52] |
| Urea | 90 | 24 | 0.335 | 1.3 | 1.89 KΩ | – | – | [41] |
| Urea | 140 | 4 | 0.355 | 1.49 | 3.72 Ω·cm | 61.40m$^2$/g | 6.53nm | This study |
| Vc | 100 | 3 | 0.354 | 0.91 | 1.82 Ω·cm | 36.88m$^2$/g | 14.98nm | This study |
| Vc+Urea | 120 | 2 | 0.348 | 0.75 | 1.2 Ω·cm | 88.60m$^2$/g | 5.72nm | This study |

those of rGO-Urea, where the sheets appear more intact. This suggests that reduction using urea introduces fewer structural defects compared to reduction using Vc. Combined with the BET results showing rGO-Urea possesses a higher specific surface area and porosity than rGO-Vc, this observation further indicates that the irregular folds generated by Vc reduction are closely linked to the extent of deoxygenation. Interestingly, rGO-Vc+Urea exhibits wrinkles similar to rGO-Vc while also retaining the more intact sheet morphology characteristic of rGO-Urea, suggesting it combines structural features of both single-agent reduced materials.

## Reduction mechanisms exploration

The primary oxygen-containing functional groups in graphene oxide comprise hydroxyl, epoxy, carbonyl, and carboxyl groups [53].Vc contains a conjugated dienol structure that readily donates hydrogen (H) atoms, conferring strong reducing capability. This property enables Vc to react with the epoxy groups, carboxyl groups, carbonyl groups, and other functional groups within GO, ultimately yielding reduced rGO. Gao et al. [29] suggested that the reduction mechanism involves a combination of S$_N$2 nucleophilic substitution and thermal reactions. Specifically, the five-membered ring in Vc can absorb electrons, facilitating the dissociation of hydroxyl groups into H$^+$ protons and alkoxide anions. These anions subsequently attack epoxy or hydroxyl group on GO, forming an intermediate that decomposes thermally to yield C=C bonds. In contrast, urea acts as a GO reducing agent primarily through thermal decomposition, generating reductive gases such as ammonia (NH$_3$) and isocyanate (NCO-). These gaseous nucleophiles remove oxygen-containing groups (e.g., carbonyl and carboxyl groups) from the GO surface. The rapid expansion of these gases creates localized mechanical stress, aiding in the separation of stacked layers [16]. Furthermore, nitrogen-containing species (e.g., amino groups, pyrrolic nitrogen [52] generated during urea decomposition react with hydroxyl and epoxy groups on GO, resulting in nitrogen-doped graphene. This doping thus enhances the material's electrical conductivity and chemical stability [54].

Comparing the reduction mechanisms of Vc and urea, both agents utilize nucleophilic attack to react with oxygen-containing functional groups in GO. Resulting intermediates are subsequently reduced to form C=C or C-N groups under thermal or other conditions, thereby restoring the sp$^2$-hybridized carbon network of graphene. The synergistic reaction pathways when Vc and urea act as co-reducing agents are proposed as follows: Urea decomposition generates an alkaline environment (Eq. (1)). This alkalinity inhibits the acid-catalyzed degradation of Vc while facilitating nucleophilic attack on GO oxygen-containing moieties, such as epoxide ring-opening.

$$(NH_2)_2\,CO + H_2O \rightarrow 2NH_3 \uparrow + CO_2 \uparrow \tag{1}$$

Vc directly donates electrons to GO through the enediol group, preferentially reducing the epoxy and hydroxyl groups (high potential matching) as in Eq. (2) [29]

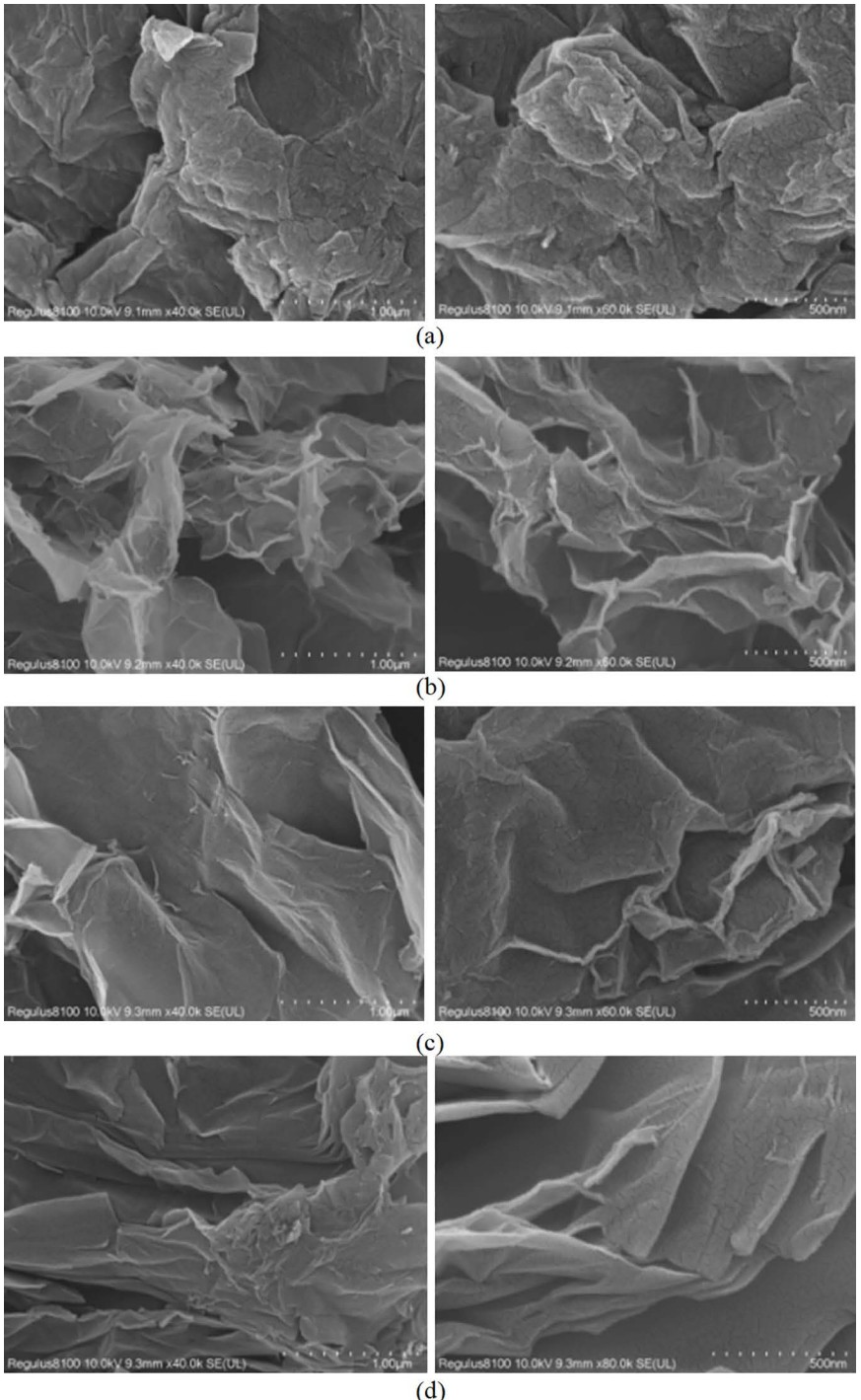

**Fig 6. SEM images of GO(a), rGO-Vc(b), rGO-Urea(c) and rGO-Vc+Urea(d) (rGO-Vc(Vc/GO mass ratio of 1 to 1,100°C,3h), rGO-Urea(Urea/GO mass ratio of 10 to 1, 140°C, 4h)and rGO-Vc+Urea (mass ratio of GO(10):Vc(0.5):Urea(10)), 120°C, 2h)).**

$$(2)$$

Partial reduction of carbonyl and carboxyl groups proceeds via a decarboxylation pathway, releasing $CO_2$. Simultaneously, $NH_3$ reacts with Vc-derived radicals: acting as a proton acceptor to facilitate Vc regeneration, while the resulting amino radical ($\cdot NH_2$) attacks oxygen-functionalized sites on GO to form C–N bonds (nitrogen doping), as detailed in Eqs. (3)–(4).

$$(3)$$

$$(4)$$

-$NH_2$ radicals undergo insertion at GO edges/vacancies to form pyridine nitrogen. Carbon radicals (R·) produced during Vc-mediated reduction combine with $NH_3$ to form amino functional groups (-$NH_2$). Vc simultaneously repairs some defects within the carbon lattice, restoring the $sp^2$-hybridized network. This dual action inhibits carbon vacancy formation, typically induced by high-temperature urea decomposition when implemented as a sole reducing agent.

CO$_2$ released during Vc-mediated reduction of carboxyl groups is sequestered by $NH_3$ (from urea hydrolysis), forming ammonium carbamate ($NH_2COONH_4$) as shown in Eq. (5). thereby lowering $CO_2$ concentration, reducing competition for Vc oxidation, protecting Vc from conversion to dehydroascorbic acid (DHA), extending the effective reduction cycle duration.

$$CO_2 + 2NH_3 \rightarrow NH_2COONH_4 \tag{5}$$

## Conclusion

In this study, rGO was synthesized using a synergistic combination of Vc and urea as co-reducing agents. Comparative analysis revealed that co-reduction significantly decreased both reaction time and temperature relative to single-agent systems (Vc or urea alone), while producing rGO with enhanced electrical conductivity (1.2 Ω·cm for rGO-Vc+Urea).

Structural characterization demonstrated rGO-Vc+Urea exhibits. Structural characterization of samples prepared with different reducing agents revealed that rGO-Vc+Urea exhibits a typical graphene structure with an interlayer spacing (d-space) of 0.348 nm and incorporates C-N structures resulting from nitrogen doping. Furthermore, rGO-Vc+Urea showed improved crystallinity order ($I_D/I_G$ 0.75) and a higher specific surface area (88.60 $m^2$/g) compared to samples reduced by single agents. Mechanistic studies indicate that the alkaline environment generated during urea decomposition potentiates Vc-mediated reduction of oxygen-functional groups (hydroxyl, epoxy, carbonyl, carboxyl) and accelerates scavenging of reaction-derived $CO_2$. These performance advantages originate from complementary reduction pathways and reactive intermediates, providing a foundational strategy for optimizing rGO synthesis for targeted applications.

## Supporting information

**S1 Table. Information about all medications.**
(DOCX)

## Author contributions

**Conceptualization:** Fei-hu Zeng.

**Data curation:** Fei-hu Zeng, Zhi-wen Wang.

**Formal analysis:** Fei-hu Zeng, Feng Wang.

**Funding acquisition:** Feng Wang.

**Investigation:** Fei-hu Zeng.

**Methodology:** Fei-hu Zeng.

**Resources:** Fei-hu Zeng, Feng Wang.

**Supervision:** SyYi Sim.

**Visualization:** Zhi-wen Wang.

**Writing – original draft:** Fei-hu Zeng, SyYi Sim.

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
