## [Decision Letter · Decision Letter 0]

28 Jul 2025

Dear Dr. FEIHU,

Thank you for submitting your manuscript to PLOS ONE. After careful consideration, we feel that it has merit but does not fully meet PLOS ONE’s publication criteria as it currently stands. Therefore, we invite you to submit a revised version of the manuscript that addresses the points raised during the review process.

We look forward to receiving your revised manuscript.

Kind regards,

Anil Kumar Reddy Police

Academic Editor

PLOS ONE

Journal Requirements:

“This work was supported by the Science and Technology Program (LT202103) and Innovation Team Program of Liming Vocational University.”

4. We are unable to open your Supporting Information file “plos one figure.rar”. Please kindly revise as necessary and re-upload.

5. We note that your Data Availability Statement is currently as follows: All relevant data are within the manuscript and its Supporting Information files.

7. Please upload a copy of Figure 1-6, to which you refer in your text on page 7-16. If the figure is no longer to be included as part of the submission please remove all reference to it within the text.

Reviewers' comments:

Reviewer's Responses to Questions

**Comments to the Author**

1. Is the manuscript technically sound, and do the data support the conclusions?

Reviewer #1: Yes

Reviewer #2: Yes

Reviewer #3: Yes

2. Has the statistical analysis been performed appropriately and rigorously?

Reviewer #1: Yes

Reviewer #2: N/A

Reviewer #3: No

3. Have the authors made all data underlying the findings in their manuscript fully available?

Reviewer #1: Yes

Reviewer #2: Yes

Reviewer #3: Yes

4. Is the manuscript presented in an intelligible fashion and written in standard English?

Reviewer #1: Yes

Reviewer #2: Yes

Reviewer #3: Yes

Reviewer #1: The manuscript presents a study on the synergistic reduction of graphene oxide (GO) using vitamin C (VC) and urea as co-reducing agents. The authors demonstrate that this combination improves reduction efficiency, lowers reaction temperature and time, and enhances the structural and electrical properties of the resulting reduced graphene oxide (rGO). The topic is timely and relevant, especially in the context of green chemistry and sustainable nanomaterials.

• The study addresses an important topic in graphene chemistry using environmentally friendly reducing agents.

• The experimental design is generally sound, and the characterization techniques are appropriate.

• The comparison between single and co-reducing agents is well-motivated and supported by data.

However, the manuscript requires major revisions before it can be considered for publication. Below are detailed comments and suggestions:

Language and Grammar:

The manuscript contains numerous grammatical and syntactical errors that hinder readability. A thorough revision by a native English speaker or professional editing service is strongly recommended. Examples:

• Pag 17 line 11: “more order” → “more ordered”

Formatting Errors:

• Equations (1) – (5) are not properly formatted or numbered in the text.

• There is a persistent formatting issue: “Error! Reference source not found.” in the BET section. This must be corrected.

Experimental Details

The manuscript lacks critical experimental parameters:

• Exact VC/GO and Urea/GO mass ratios. Mass ratios for rGO-VC and rGO-Urea are omitted in Section 2.2. For rGO-VC+Urea, "mass ratio of rGO(10):Vc(0.5):Urea(10)" is confusing (rGO is the product). Specify as GO:VC:urea = 10:0.5:10.

• pH of the reaction mixtures.

• Details on the reproducibility of the synthesis (e.g., number of replicates).

Data Presentation

Abstract/Results: Claims "superior electrical conductivity (1.94 Ω)" are incorrect.

• Ω (ohm) measures resistance, not conductivity (measured in S/cm or S/m).

Table 1 Issues:

• Units for "rGO conductivity" are mixed (Ω, S·cm⁻¹, F·g⁻¹). Include a column for metric type (e.g., resistance, conductivity, capacitance).

• The BET data should be summarized in a table with surface area, pore volume, and average pore size.

Discussion

• The proposed mechanism is interesting but speculative. The authors should support their claims with additional evidence (e.g., XPS for nitrogen doping).

• The role of urea in creating an alkaline environment and its interaction with VC is plausible but needs clearer chemical justification.

References:

• Ref 33 (Ren et al., 2025) is post-submission (2025). Verify accessibility/cite as "in press" if applicable.

• Ref 21: Author list truncated ("B.Lesiak, et al.").

• Some inconsistencies in formatting and citation numbering. Please revise

The manuscript has potential but requires substantial improvements in language, formatting, and data presentation. I encourage the authors to address the points above carefully and resubmit a revised version.

Reviewer #2: The work is scientifically sound, and the manuscript is generally well-drafted. However, the following points should be addressed to enhance the quality and clarity of the manuscript:

1. In the Materials and Chemicals section, please list all the chemicals used along with their respective purities.

2. In the rGO Preparation section, clearly specify the mass ratios of the reagents, reaction temperatures, and durations involved in the synthesis process.

3. There is a lack of experimental data for the rGO-VC+Urea sample in terms of resistivity and resistance. Please include more rGO-VC+Urea samples to allow proper comparison with the rGO-VC and rGO-Urea samples.

4. The Y-axis title is missing in the XRD figure.

5. The phrase “more order” in the statement “Compared to rGO-Urea, the XRD peaks of rGO-VC and rGO-VC+Urea are more order” is unclear. Please elaborate on what is meant by “more order” — for example, whether it refers to higher crystallinity, or sharper peaks.

6. Remove the text “Error! Reference source not found.” in the BET analysis section and provide the correct reference to support the statement.

7. In Table 3, please provide appropriate references for Urea, VC, and VC+Urea. For samples obtained in the current work, label them as “this study.”

Reviewer #3: The authors in the present manuscript show the rGO (rGO-VC+Urea) was prepared by combining vitamin C (VC) and urea as co-reducing agents with the modified Hummer’s method. Compared to samples reduced solely with VC or urea, the co-reducing agents significantly reduced the required reaction time (to 2 hours) and temperature (to 120°C), while yielding material with superior electrical conductivity (1.94 Ω). The structure of the samples was characterized using XRD, FT-IR, Raman spectroscopy, BET surface area analysis, and SEM. Results indicate that the sample prepared from co-reducing agents possesses a typical graphene structure and incorporates C-N bonds. The authors should address the following issues and information’s before publication acceptance in the prestigious ‘PLOS One’ Journal:

1. In Introduction, authors should add a Table that compares the rGO, preparation methods and physicochemical characteristics with published literatures?

2. On what basis do authors decide to use vitamin C and urea for reduction of GO?

3. In Introduction, authors should explain deeply what is the novelty of this study?

4. In Introduction section, authors should add a short paragraph on characteristics of graphene oxide (GO). Authors may go through these publications for more details and cite accordingly: https://doi.org/10.1016/j.cartre.2023.100251 & https://doi.org/10.1016/j.carbon.2024.119331

5. In rGO Preparation, authors should correct the spelling “Preparation”? What is the impact of ultrasonication on GO?

6. In BET analysis, authors should explain why the surface area is highest for rGO prepared from combination of VC and urea?

7. In XRD spectra, two peaks around 11° and 13° can be observed for rGO prepared from urea, what are these peaks?

8. Authors may shift main figures into the manuscript instead of presenting them in supplementary information.

**Do you want your identity to be public for this peer review?** For information about this choice, including consent withdrawal, please see our Privacy Policy

Reviewer #1: No

Reviewer #2: No

Reviewer #3: No

---

## [Author Response · Author response to Decision Letter 1]

5 Aug 2025

1. All revisions requested by the editors have been addressed in the updated manuscript.

2. The manuscript has undergone professional language editing by Sim Syyi, ensuring corrections of usage, spelling, and grammar. The following files are resubmitted:

- Response to Reviewers

- Revised Manuscript with Track Changes

- Clean Manuscript

- Supporting Information

3. The funding disclosure statement:

has been added to the new cover letter.

4. All figures have been re-uploaded individually in the specified format.

5. Complete experimental datasets are provided under "Other" in Supporting Information.

---

## [Decision Letter · Decision Letter 1]

11 Aug 2025

Synergistic Reduction of Graphene Oxide Using Vitamin C and Urea: Enhanced Efficiency and Material Properties

PONE-D-25-36950R1

Dear Dr. FEIHU,

We’re pleased to inform you that your manuscript has been judged scientifically suitable for publication and will be formally accepted for publication once it meets all outstanding technical requirements.

Kind regards,

Anil Kumar Reddy Police

Academic Editor

PLOS ONE

Additional Editor Comments (optional):

Reviewers' comments:

Reviewer's Responses to Questions

**Comments to the Author**

Reviewer #1: All comments have been addressed

Reviewer #2: All comments have been addressed

Reviewer #3: All comments have been addressed

2. Is the manuscript technically sound, and do the data support the conclusions?

Reviewer #1: Yes

Reviewer #2: Yes

Reviewer #3: Yes

3. Has the statistical analysis been performed appropriately and rigorously?

Reviewer #1: Yes

Reviewer #2: Yes

Reviewer #3: Yes

4. Have the authors made all data underlying the findings in their manuscript fully available?

Reviewer #1: Yes

Reviewer #2: Yes

Reviewer #3: Yes

5. Is the manuscript presented in an intelligible fashion and written in standard English?

Reviewer #1: Yes

Reviewer #2: Yes

Reviewer #3: Yes

Reviewer #1: The authors have addressed all the issues raised by making the appropriate changes to the manuscript. The manuscript can be published in its current form.

Reviewer #2: (No Response)

Reviewer #3: (No Response)

**Do you want your identity to be public for this peer review?** For information about this choice, including consent withdrawal, please see our Privacy Policy

Reviewer #1: **Yes: ** Angela Longo

Reviewer #2: No

Reviewer #3: No

---

## [Editor Report · Acceptance letter]

PONE-D-25-36950R1

PLOS ONE

Dear Dr. Zeng,

I'm pleased to inform you that your manuscript has been deemed suitable for publication in PLOS ONE. Congratulations! Your manuscript is now being handed over to our production team.

Kind regards,

on behalf of

Dr. Anil Kumar Reddy Police

Academic Editor

PLOS ONE